biomaterials/neuroscience/structural biology

neural networks, three-dimensional structuring, polymer particles, small-world, electrophysiology, connectivity

**Author for correspondence:**
Ioanna Sandvig
e-mail: ioanna.sandvig@ntnu.no

# Formation of neural networks with structural and functional features consistent with small-world network topology on surface-grafted polymer particles

Vibeke Devold Valderhaug[1], Wilhelm Robert Glomm[2], Eugenia Mariana Sandru[2], Masahiro Yasuda[3], Axel Sandvig[1,4,5] and Ioanna Sandvig[1]

[1]Department of Neuromedicine and Movement Science, Faculty of Medicine, Norwegian University of Science and Technology (NTNU), 7030 Trondheim, Norway
[2]SINTEF AS, Department of Biotechnology and Nanomedicine, Trondheim, Norway
[3]Department of Chemical Engineering, Osaka Prefecture University, 1-1 Gakuen-cho, Naka-ku, Sakai, Osaka 599-8531, Japan
[4]Department of Neurology and Clinical Neurophysiology, St Olav's Hospital, Trondheim, Norway
[5]Department of Pharmacology and Clinical Neuroscience, Division of Neuro, Head and Neck, Umeå University Hospital, Umeå, Sweden

VDV, 0000-0001-5523-0725

*In vitro* electrophysiological investigation of neural activity at a network level holds tremendous potential for elucidating underlying features of brain function (and dysfunction). In standard neural network modelling systems, however, the fundamental three-dimensional (3D) character of the brain is a largely disregarded feature. This widely applied neuroscientific strategy affects several aspects of the structure–function relationships of the resulting networks, altering network connectivity and topology, ultimately reducing the translatability of the results obtained. As these model systems increase in popularity, it becomes imperative that they capture, as accurately as possible, fundamental features of neural networks in the brain, such as small-worldness. In this report, we combine *in vitro* neural cell culture with a biologically compatible scaffolding substrate, surface-grafted polymer particles (PPs), to develop neural networks with 3D topology. Furthermore, we investigate their electrophysiological network activity through the use of 3D multielectrode arrays. The resulting neural network activity shows emergent behaviour

consistent with maturing neural networks capable of performing computations, i.e. activity patterns suggestive of both information segregation (desynchronized single spikes and local bursts) and information integration (network spikes). Importantly, we demonstrate that the resulting PP-structured neural networks show both structural and functional features consistent with small-world network topology.

## 1. Introduction

Combining *in vitro* neural network models with tools for electrophysiological investigation is an established (modelling) approach for exploring the emerging activity and function of neural networks. Recent advances in morphogenetic neuroengineering have led to a surge of scientific interest aimed at using these already established tools in novel ways. The well-established, standard neural network modelling approach has been to create monolayer neural networks from dissociated neural tissue or from neural stem cells, and to measure the emerging network activity using microelectrode arrays (MEAs). Some fundamental traits of brain networks, such as self-organization and spontaneous network formation and activity, are recapitulated by these models, making them attractive reductionist paradigms for neuroscientific research. Some evidence, however, points towards a prominent activity feature emerging in these *in vitro* neural networks that is largely incompatible with the activity of the brain, namely highly synchronized activity [1–3]. This discrepancy limits the relatability and thus the potential information that can be gained from this otherwise valuable approach. Knowledge gained in the field of connectomics, however, suggests that this limitation can be overcome. A highly interdependent nature of structure and function in the neural networks of the brain has been uncovered [4–6], which implies that a more realistic topology may need to be recapitulated in our standard modelling systems if they are to produce networks with activity and function traits more relatable to those seen in the brain.

The pattern of physical interconnections and the activity of a neural network are critically interdependent, where the strength and directness of the physical interconnections between neuronal ensembles have been shown to determine and constrain their functional interactions [4,5]. Several attempts at structuring *in vitro* neural networks have therefore been reported [7], as standard monolayer culture mainly allows connections to form in one plane ($X,Y$), disregarding the third ($Z$) dimension which greatly influences the structure of biological neural networks. A few of these studies have also compared the electrophysiological network activity of monolayer neural networks and neural networks structured in three dimensions (3D) using standard two-dimensional (2D) MEAs, where the results indicate an effect on global network synchrony and random spiking due to the structuring [1,8]. Further supporting this strategy is the small-world topology of the brain, a characteristic feature which facilitates the simultaneous capacity of information integration and segregation, the two emerging network phenomena recognized as the basis of behaviour [2,5,9,10]. Computational functions which are spatially and temporally segregated into functional modules in the brain are dynamically engaged and disengaged through transient phase or frequency locking, i.e. oscillations/ synchronization, and thus integrated into transitory coordinated global functions [2]. Furthermore, the small-world topology, characterized through its simultaneous high clustering and characteristic short path lengths [5,11], has been shown to facilitate the spread of disease to a greater extent than other network architectures [12–15]. This might relate to the development of neurodegenerative disorders such as Alzheimer's and Parkinson's disease, or dementia with Lewy body pathology, which have all been hypothesized to progress through the propagation of pathological protein aggregates through interconnected brain areas [16,17]. Thus, basic features of the human brain connectome, such as small-worldness and internodal connectedness, influence both network function and dysfunction, highlighting the fundamental importance of capturing these features in our modelling systems.

In this report, we show that we can capture some of the complexity of neural networks in the brain through interfacing *in vitro* neural cell cultures with surface-grafted, non-conducting, polymer particles (PPs) to create neural networks with 3D topology. Previously, these PPs have been successfully employed as microenvironments for creating 3D bone marrow culture systems, which have been used for both haematopoietic stem cell studies [18–20] and chemosensitivity studies [21]. In the present study, we report the structuring of neural networks using PPs combined with 3D MEAs for electrophysiological network measurements and show how the resulting structural and functional network traits relate to a small-world network topology.

# 2. Material and methods

## 2.1. Fabrication of polymer particles with surface-grafted chains

Poly(vinyl pyrrolidone) K90 (PVP) of average molecular weight approximately $360\,000\,g\,mol^{-1}$, pentaerythritol triacrylate (PETA), methacrylic acid (MA) glycidyl methacrylate (GMA) and toluene were purchased from Sigma Aldrich. Methyl methacrylate (MMA) was purchased from Fluka. 2,2'-Azobis(isobutyronitrile) (AIBN) was obtained from Akzo Nobel. 2,2'-azobis[N-(2-propenyl)-2-methylpropionamide] (APMPA) was obtained from Wako Pure Chemical Co. (Osaka, Japan). All reagents were used without further purification.

### 2.1.1. Synthesis of PPs

PPs with surface-grafted epoxy-containing polymer chains were synthesized via suspension polymerization using a protocol adapted from Yasuda et al. [20]. Briefly, 56 ml of an aqueous 2% PVP solution was added to a 100 ml temperature-controlled glass reactor with an impeller, and stirred (500 r.p.m.) at 25°C. In a separate vessel, 0.2 g of AIBN, 0.3 g of APMPA, 3.8 g of MMA and 3.8 g of PETA were mixed and the resulting monomer mixture was added dropwise to the reactor under stirring (500 r.p.m.). Polymerization was done in the reactor first at 70°C for 3 h and then at 80°C for 2 h under stirring (350 r.p.m.). Following synthesis, the particles were washed three times in DI water and dried using azeotropic distillation prior to further functionalization. Particle size distribution was analysed via optical microscopy and laser diffraction, and ranged from 100 to 1000 µm, with a volume average around 300 µm. The stability of the resulting surface-grafted PPs in cell culture medium (RPMI 1640 with 10% FBS, 1% L-Glut and 1% PenStrep (1 : 1 penicillin/streptomycin)) was subsequently investigated, revealing a 54% reduction in volume average diameter after 24 days.

### 2.1.2. Functionalization of PPs

Surface grafting of epoxy-containing copolymer chains was done using a protocol adapted from Yasuda et al. [20]. In a typical procedure, 5 g of PPs, 12.5 g of GMA and 3.1 g of MA were added to a three-neck flask together with 125 g of toluene. The grafting reaction was carried out for 8 h at 105°C under stirring (100 r.p.m.), and the reaction mixture was subsequently gradually cooled down to room temperature. After filtration, the particles were washed three times with ethanol and three times with water and stored in ethanol at 4°C until use in cell cultures.

## 2.2. Establishment of 3D neural networks on PPs

Rat fetal neural stem cells (NSCs) (Gibco) were seeded onto CellStart® (Gibco) coated vessels and maintained in Complete Stem Pro® NSC SFM media (Gibco) supplemented with 1% penicillin–streptomycin. For seeding together with the PPs, the media were further supplemented with 1% BSA, 0.1% ROCK inhibitor, and 0.5 µg ml$^{-1}$ fibronectin.

The PPs were transferred with a spatula to 1.5 ml Eppendorf tubes (about 200 µl of dry PPs in each tube), where they were washed three times in PBS, and three times in PBS containing 10% FBS. The supernatant was then removed, and an equivalent amount of cell suspension (200 µl containing $2 \times 10^5$ NSCs) was added to each tube and mixed gently. The mixture was then transferred to the relevant culture vessel and incubated in 37°C for 1 h before more media were added to the NSC cultures with PPs. Differentiation towards neural lineage was initiated 2 days post-seeding through the use of a differentiation medium consisting of Neurobasal supplemented with 2% B27, 1% GlutaMAX and 1% penicillin–streptomycin.

### 2.2.1. Microelectrode array preparation

Sixty-electrode 3D MEAs (60-3DMEA200/12/50iR-Ti; Multichannel Systems) with ring covers were used as culture vessels for cell seeding and for recording spontaneous electrophysiological activity of the developing neural networks on the PPs. Recordings were obtained through the MEA2100 in vitro system and suite software (Multi Channel Systems; Reutlingen, Germany). In addition, seeding rings (MEA ALA-inserts; Multi Channel Systems; Reutlingen, Germany) were used to reduce movement and keep the PPs in place over the electrodes throughout the experimental period.

Before seeding, the 3D MEAs were washed with 65% ethanol, incubated in sterile water and UV-treated. To make the culture surface hydrophilic, the MEAs were subsequently treated with fetal bovine serum (FBS) for 60 min.

Prior to seeding on 3D MEAs, the NSCs were labelled with a carbocyanine lipophilic tracer, DilC18 (5) DiD (L7781, Invitrogen) (excitation 644 nm). A 5 µl cell-labelling solution was mixed into 1 ml cell suspension containing $1 \times 10^6$ rat NSCs by gentle pipetting. The resulting solution was then incubated for 20 min at 37°C, washed and centrifuged three times at $200g$ for 3 min. The cells were then mixed with the PPs as described above and transferred into the seeding ring on the 3D MEAs using a spatula. An additional 100 µl of cell suspension was added once the mixture was in place within the seeding ring. The resulting NSC cultures with PPs on 3D MEAs were then incubated at 37°C for 1 h for the cells to adhere, before another 350 µl of media were added. The NSC were differentiated and the derived 3D neural networks on the PPs were maintained on the 3D MEAs for three to four weeks post-seeding, throughout which fluorescence microscopy and electrophysiological recordings were obtained.

### 2.2.2. Immunocytochemistry of neural networks on PPs

For fixation, 2% paraformaldehyde (PFA) was added to the 3D neural networks on the PPs for 10 min, followed by 4% PFA for 10 min and two 15 min washes in PBS. This was followed by a 1 h incubation in blocking solution consisting of PBS with 5% normal goat serum and 0.3% Triton-X, and an overnight incubation in primary antibody solution (PBS containing 1% normal goat serum, 0.1% Triton-X) at 4°C. The following primary antibodies from Abcam were used: rabbit anti-CaMK2 (1 : 400), rabbit anti-Synaptophysin (1 : 250), rabbit anti-GFAP antibody (1 : 1000), mouse anti-CNPase antibody (1 : 500), mouse anti-PSD95 (1 : 300), mouse anti-β 3 tubulin (1 : 1000) and chicken anti-MAP2 antibody (1 : 1000), chicken anti-neurofilament heavy (1 : 500) and chicken anti-GFAP antibody (1 : 1000). The neural networks on the PPs were then washed twice for 15 min in PBS and incubated in secondary antibody solution (PBS containing 1% normal goat serum, 0.1% Triton-X) in the dark, at room temperature, with the following secondary Alexa Fluor™ 568, 488 and 647 antibodies (Thermo Fisher, MA, USA) at a dilution of 1 : 1000. Hoechst (1 : 10 000) was added for the final 5 min before another two 15 min wash in PBS was conducted. Images were taken using a Zeiss Axiovert 1A fluorescent microscope (Carl Zeiss, Germany) and Zen 2.3 Lite, Blue Ed. Software. ImageJ and PowerPoint were used to post-process the images.

### 2.2.3. Investigation of structural topology by scanning electron microscopy of neural networks on PPs

Samples of the 3D neural networks on the PPs were washed twice in PBS before being fixed in a solution of 2.5% glutaraldehyde with 2% PFA in 0.1 M Hepes buffer for 3 h at room temperature, followed by overnight fixation in 4°C. The samples were then washed twice for 5 min in Hepes buffer, subsequently dehydrated in 5 min steps using increasing ethanol concentrations (20–50–70–90–100–100% ethanol), and dried in 10 min steps using hecamethyldisiloxane (HMDS) (50% and 2× 100%) before being air-dried in a desiccator. The samples were subsequently mounted on aluminium pins with double-sided carbon tape and sputter coated (Polaron) with Gold/Palladium (30 nm thickness). The samples were then examined using a scanning electron microscope (SEM) (Teneo SEM, Thermo Fisher Scientific) at 10–15 kV with an ETD detector.

### 2.2.4. MEA recordings and data analysis

Recordings were obtained through the MEA2100 *in vitro* head stage, interface system and suite software (Multi Channel Systems; Reutlingen, Germany), and are available within the Mendeley data repository (http://dx.doi.org/10.17632/r9gd7g8zcy.4 [22]). Samples of the electrophysiological activity of the 3D neural networks on the PPs were recorded for durations of approximately 90 consecutive seconds throughout the experimental period, with a sampling rate of 10 kHz, where a waveform amplitude exceeding a threshold of ±5 s.d. from the mean was registered as a spike. An in-house developed MEA data analysis Toolbox (available for download at https://github.com/helgeanl/MEA_toolbox) was used to visualize the waveforms of each spike, and activity raster plots were produced based on the timestamp of each spike. After the final electrophysiological recording, the media were replaced with PBS, and the 3D neural networks were left to dry on the 3D MEAs at room temperature, followed by an overnight UV-treatment to terminate the cells. Recordings were subsequently obtained from these 3D MEAs as a control to test whether the PPs themselves produced any electrophysiological artefacts. In addition, impedance measurements were made to test whether any damage had been done to the electrodes.

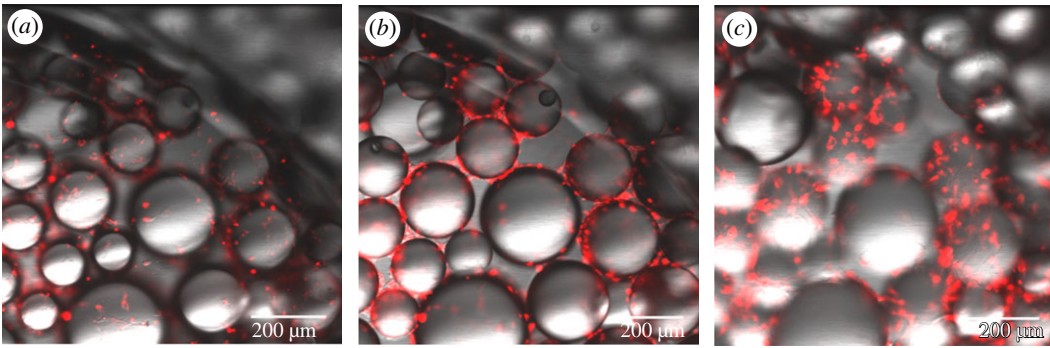

**Figure 1.** Confocal images demonstrating the emergence of 3D neural networks on the surface-grafted PPs 4 days after seeding, with different z-planes showing the fluorescently labelled cells (red) attached at different levels/heights of the same PPs (brightfield) (10X). (a) The bottommost focus level with fluorescently labelled cells attached to the well plate as well as the bottom of the PPs. (b) The fluorescently labelled cells about 100 µm further up in the z-plane, and (c) 200 µm further above that.

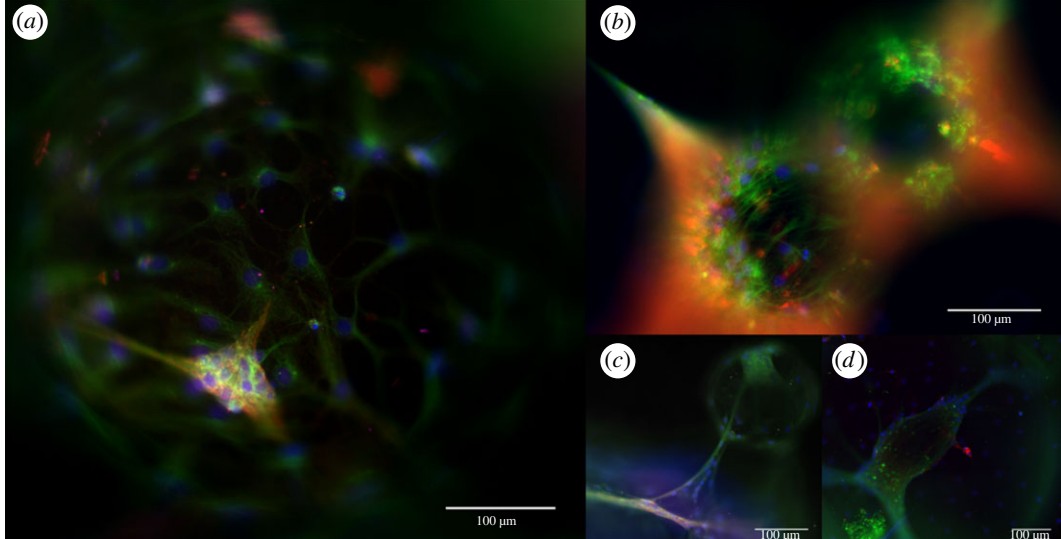

**Figure 2.** ICC of neural networks on surface-grafted PPs. Following one month of differentiation and maintenance of the neural networks on the PPs, ICC confirmed the presence of mature cells attached to the surface of the PPs. The images show neural networks expressing fluorescently labelled markers for (a) calcium/calmodulin protein-dependent kinase-II (CAMK2) (green), post-synaptic densities-95 (PSD95) (red) and nuclear marker Hoechst (blue), (b) for synaptophysin (red), GFAP (astrocytic marker) (green) and Hoechst (blue), (c) for PSD95 (red), CAMK2 (green), neurofilament heavy (grey) and Hoechst (blue) and (d) for synaptophysin (red), GFAP (green) and Hoechst (blue). Scale bar, 100 µm.

## 3. Results

### 3.1. Neural network culture

Rat NSCs were successfully seeded and maintained among the PPs. The resulting neural networks could be imaged with fluorescence microscopy, showing cells labelled with a lipophilic tracer growing underneath, around and on top of the PPs as early as 4 days post-seeding (figure 1). Although we contend with some loss of cells with each media change, we were able to maintain the neural cell cultures on the PPs for over one month. After this point, the cultures were fixed and immunolabelled with neural lineage markers. The immunocytochemistry (ICC) confirmed the presence of neurons (MAP2[+]), as well as astrocytes (GFAP[+]) and oligodendrocytes (CNPase[+]) attached to the surface of the PPs after one month in culture (electronic supplementary material, figure S1). Furthermore, the neural networks were positively immunolabelled with markers for mature axons (neurofilament heavy polypeptide) as well as synaptic vesicles (synaptophysin), post-synaptic elements (PSD95), calcium/calmodulin protein-dependent kinase-II (CAMK2), which is involved in neurotransmitter secretion, synaptic connectivity and long-term potentiation, and GFAP (figure 2).

Based on standard morphological characteristics as well as careful comparison with the demonstrated ICC results, topographical SEM investigation confirmed that the PPs did indeed support the

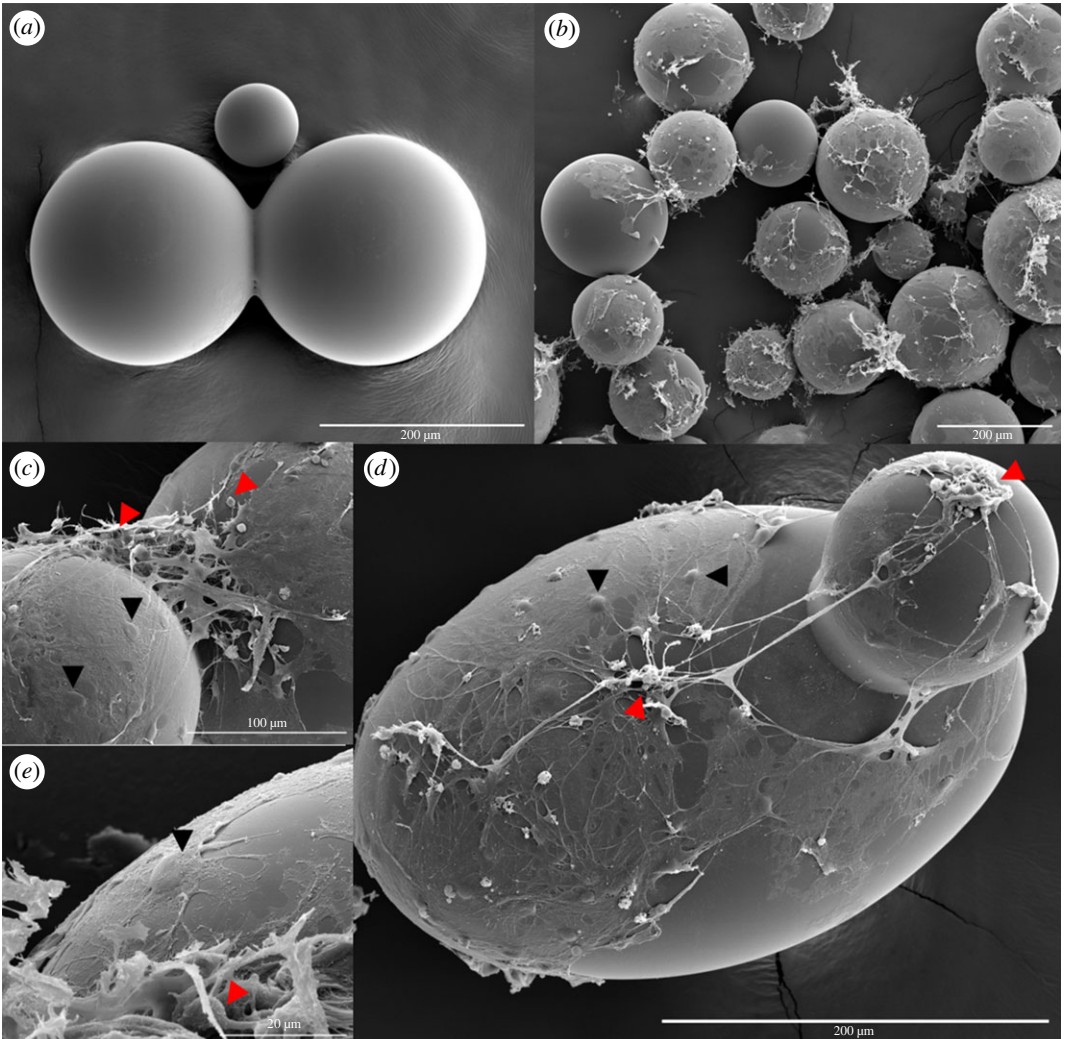

**Figure 3.** Prominent 3D neural network formation on surface-grafted PPs. Topography investigation by SEM confirmed the presence of extended neural networks with 3D topology on the surface of, and connecting between, the PPs three weeks post-seeding. (*a*) SEM image displaying the PPs alone (200 μm scale bar) and (*b*) after establishment of 3D neural networks (200 μm scale bar). (*c*) Higher magnification image of the intricate 3D neural network connections spanning the gap between two PPs (100 μm scale bar). Neuronal cell bodies (red arrowhead) with axonal projections can be distinguished within the network, as well as glial cell bodies (black arrowhead) with thin membranous extensions covering the particle surface. (*d*) Image revealing clusters of neurons (red arrowhead) connected through suspended axonal bundles on different PPs. Glial cell bodies (black arrowheads) can be distinguished regularly tiling most of the particle surface (200 μm scale bar). (*e*) Detailed image displaying another neuronal cell body (red arrowhead) with thin neurites and a thicker axonal projection, as well as another glial cell (black arrowhead) covering the surface with a thin membranous sheath (20 μm scale bar).

establishment of prominent 3D neural networks (figure 3). These networks covered the surface of the PPs as well as interlaced among them. Furthermore, glial cell bodies were observed regularly tiling the surface and wrapping most of the PPs with thin membranous extensions, while neuronal clusters connected through axonal projections could be observed on top. Importantly, several suspended axon bundles interconnecting neuronal clusters on different PPs could be observed, providing structural 'short-cut' connections between neuronal clusters located at different levels in the network (figure 3*d*).

## 3.2. Electrophysiological measurements of the developing 3D neural networks on the PPs using 3D MEAs

Three-dimensional neural networks on the PPs were successfully developed and maintained on 3D MEAs for a period of three to four weeks. A lipophilic tracer used to label the rat NSCs prior to seeding and differentiation on the 3D MEAs made it possible to visualize some of the cells among the PPs during

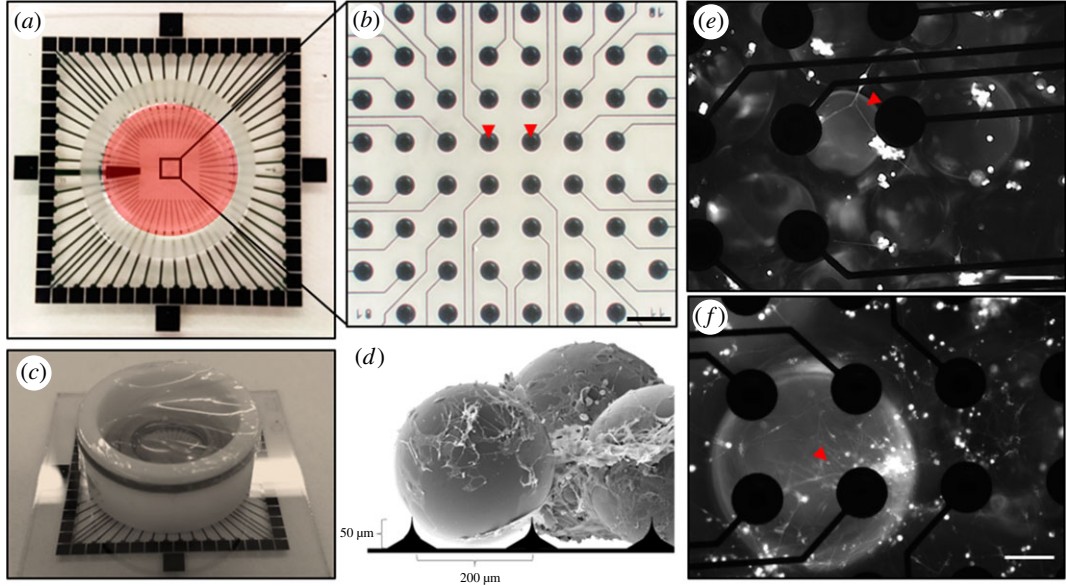

**Figure 4.** Three-dimensional MEA set-up for electrophysiological recording of the developing 3D neural networks on surface-grafted PPs. (*a*) Image of a 3D MEA. The cell culture chamber is highlighted in red. The recording area containing the electrodes are boxed in and displayed magnified in (*b*), where the red arrowheads indicate two of the 60 recording electrodes (200 µm scale bar). (*c*) Sideview of the same 3D MEA, with a ring cover protecting the 3D neural networks on the PPs from contamination during recordings. (*d*) Cartoon illustrating the PP structured neural networks together with the 3D electrodes on the MEAs. (*e*) Image displaying developing neural networks labelled with a fluorescent lipophilic tracer on the PPs 2 days after seeding on the 3D MEAs. The red arrowhead indicates neurites connecting cell clusters (100 µm scale bar). (*f*) Image displaying the fluorescently labelled developing neural networks 3 days after seeding on the 3D MEAs. The red arrowhead indicates weakly labelled neurites extending from a cell cluster centred around a single 3D recording electrode (100 µm scale bar).

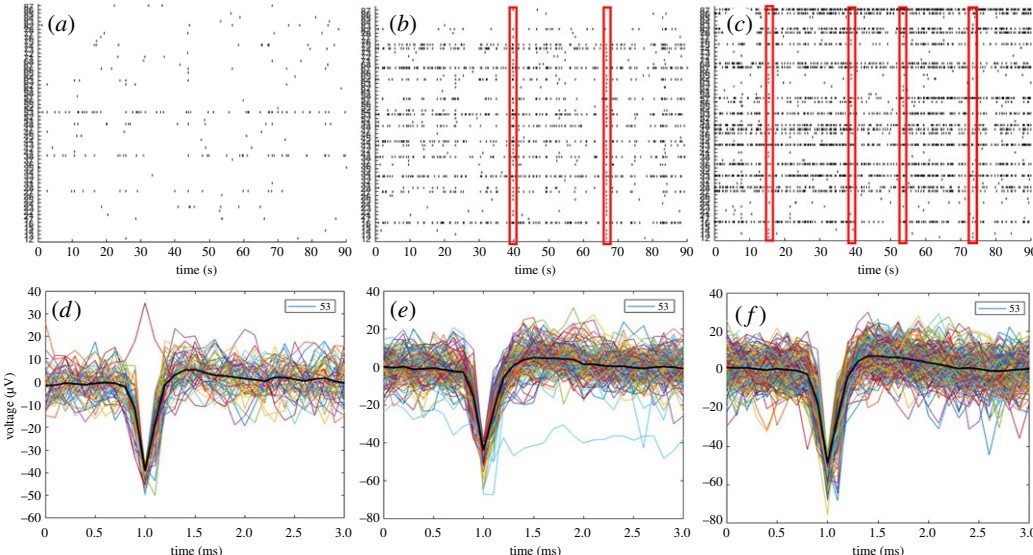

**Figure 5.** Three-dimensional developing neural network activity on surface-grafted PPs as measured from 3D MEAs. (*a*–*c*) Activity maps (raster plots) from a 3D neural network on an MEA ($n = 4$) at one, two and three weeks post-seeding, respectively. Each spike recorded during a 90 s sampling period is indicated for each of the 60 electrodes (numbered on the *y*-axis). Network spikes, events in which a spike or several spikes are detected at most of the active electrodes in the network at the same time, are highlighted in red. (*d*–*f*) The spike shape of each recorded spike at electrode 53 for each of the three timepoints (*x*-axis = ms, *y*-axis = µV).

the initial culture period (figure 4). Electrophysiological recordings performed throughout a period of three to four weeks demonstrated the emergence of electrically active, developing 3D neural networks on the PPs as shown by the activity raster plots of a representative neural network in figure 5*a*–*c*. The recordings performed during the first week showed sporadic, largely unsynchronized spontaneous action potentials scattered among the electrodes. Sample recordings performed during the second week of development demonstrated the presence of maturing networks, as patterns of more regular spiking and bursting

behaviour emerged. Indeed, even some network spikes (synchronization), i.e. single or multiple spikes detectable by most of the active electrodes in the culture within the same time period, were observed. During the third, final week of recordings, transient, regular, local bursting behaviour as well network spikes interspaced with longer periods of unsynchronized, scattered action potentials typified the electrical activity of the 3D neural networks on the PPs ($n = 4$). Furthermore, the shape of each recorded spike at one of the most active electrodes (electrode 53) is displayed for each of the three timepoints (figure 5d–f). By contrast, as a control measure, no electrical activity exceeding the threshold for noise was observed with the PPs alone, and no indication of damage to the electrodes could be read from impedance measurements or visual inspection.

## 4. Discussion

Several studies employing various scaffolds for 3D neural tissue culture have attempted to capture the basic structural dimensionality of neural networks in the brain [7]. However, very few of these studies provide electrophysiological measurements of the resulting network activity, and those that do use standard planar (2D) MEAs for this purpose [1,8]. Since the 3D MEA electrodes used in this study can measure electrical activity up to 50 µm away from a cell body along the entire surface area of the electrode, they cover a much greater area than the electrodes on conventional 2D MEAs, and are as such better suited to obtaining electrophysiological activity originating from several levels of the neural network, rather than mainly from the bottommost layers.

Two novel findings have been described in this report: firstly, we show that the presented PPs can function as long-term scaffolds for 3D neural network structuring, as they allow the attachment, survival, differentiation and maturation of neural networks for over one month (figures 2, 3 and 5; electronic supplementary material, S1). Differentiation and maturation were demonstrated with ICC through the presence of neurons with mature axons containing synaptic vesicles, post-synaptic densities and proteins involved in long-term potentiation. Furthermore, suspended neuronal connections interlacing between the PPs (figure 3c,d), connecting remote neuronal clusters on distant and otherwise independent surface areas at different levels demonstrated the 3D of the structured neural networks. These structural 'short-cuts' provide a much quicker path between the clusters than what would be possible if the connections were confined to the surface, as in standard monolayer neural networks, and provide physical connectivity features consistent with small-world network topology. Secondly, we demonstrate through electrophysiological measurements that these PP-structured 3D neural networks are functional, i.e. that they spontaneously develop emergent behaviour consistent with maturing neural networks capable of performing computations through activity patterns suggestive of information segregation (desynchronized spikes and local bursts) and information integration (network spikes) (figure 5). Furthermore, some emergent activity features consistent with a developing small-world topology were revealed in the final week of electrophysiological recordings, where the 3D neural networks displayed higher local activity clustering interspaced with a few synchronized network events.

Previously, similar PPs have been used as scaffolds for growing a range of non-neural cell types, such as fibroblasts, osteoblasts and chondrocytes, as well as for growing co-cultures of MS-5 stromal cells and HeLa cells, haematopoietic stem cell and leukaemic cells [18–21]. However, these cell types are much more robust and less sensitive to mechanical stress or fluctuations in environmental parameters such as temperature and pH compared to cells of a neural lineage. It is therefore an important finding that the PPs support the development and survival of *in vitro* neural networks as well.

In the present study, the gain of dimensionality obtained through structuring the neural networks added complexity to the network connectomes, particularly in relation to the possible 'shortest path length' of the structural connections between distant neural clusters/nodes. This can be readily observed by studying the physical interconnections preserved in the SEM preparation displayed in figure 3d, where a suspended axonal bundle spans the gap between two PPs and interconnects the neuronal clusters found on each of them. This suspension provides the shortest possible path between the remote neuronal clusters, reducing the topographical distance and processing length between the connected neuronal clusters/nodes, which is a direct result of the structuring. A similar gain of 'directness' in the connectivity between neuronal ensembles located on otherwise independent surface areas can be observed in figure 3c. This directly observable structural connectivity trait of local neural clusters interconnected with other distant neural clusters through a few axon bundles is highly consistent with a small-world network topology [11,14,15]. Furthermore, these features can determine the possible functional interactions of the neuronal clusters/nodes and the overall efficiency of the 3D

neural networks [23], as the shortest path between interconnected nodes in a network has implications for the signal propagation speed, computational power and synchronizability [11].

The electrophysiological activity of an *in vitro* neural network should reflect basic emergent phenomena of the brain, namely the simultaneous capacity of information integration and segregation [2,5,9,10]. As can be seen from the raster plots in figure 5 showing electrophysiological development, the 3D neural networks which emerge among the PPs during the second and third week post-seeding display simple forms of both unsynchronized, local bursting behaviour, consistent with segregation, as well as single network spikes/ bursts transiently engaging most of the active nodes in the network, suggestive of integration. However, these electrophysiological neural network traits are also present in *in vitro* monolayer neural networks and represent basic emergent behaviour of neural networks in general [24,25]. Nonetheless, in contrast with the highly synchronized network activity often observed in *in vitro* monolayer neural networks, largely desynchronized and local/clustered network activity was observed during the final weeks of development recorded of the PP-structured neural networks in this study. Together with the observed structural connectivity showing distant neuronal clusters connected through 'short-cut' suspended connections, these features of higher activity clustering interspaced with a few synchronized network events suggest that structuring the neural networks with the PPs facilitates the establishment of small-world topology, both at a structural and functional level.

This initial study has pointed towards key topological features consistent with small-world topology of the PP-structured neural networks, which are more in line with the *in vivo* reality than classical monolayer neural networks. Nevertheless, further investigations and optimizations are needed before a conclusion can be made about the utility of this platform. These particular PPs allow for tuning of the cell–surface interaction via the length and chemical composition of the surface-grafted chains, as well as the available volume for cell growth and connectivity, via the particle size and polydispersity, which could be further optimized for neuronal cultures. Furthermore, the biological relevance of the PPs could be increased by harnessing their capacity for graded compound release through incorporation of programmable degradability. Combined with other *in vitro* platforms, such as microfluidic chips, this could be used to model, for instance, the effect of neuroprotective compounds/new drugs on network degradation in relation to neurodegenerative diseases in a more biologically relevant manner than what is possible at the moment.

# 5. Conclusion

*In vitro* models for inferring neural activity at a population/network level hold tremendous potential for elucidating underlying features of neural network function in healthy and perturbed condition. It is therefore imperative that the basic characteristics of these widely applied *in vitro* neural network models capture fundamental structural and functional features of neural networks in the brain as accurately as possible. As we have shown, the application of neural interfaces such as the PPs presented in this report has the potential of recapitulating an important aspect of self-organization and connectivity, namely the 3D character of biological neural networks. Importantly, we have shown that the PP structuring increases the possible connectedness between remote, local neuronal clusters through suspended axon bundles, i.e. 'structural short-cuts', this paradigm has the capacity of more realistically capturing features of the small-world architecture of the brain, an attribute which can be pivotal for elucidating structure–function mechanisms translatable to the actual functional (or dysfunctional) human brain from *in vitro* neural network models. Furthermore, although we show features consistent with both structural and functional small-world topology in the PP-structured neural networks in this initial study, further investigations and optimizations of the PPs are needed before a definitive conclusion can be made about the utility of this platform in neural network modelling.

# 6. Limitations

The electrophysiological data were not post-processed, as the in-house built toolbox for MEA analysis does not allow for filtering of the recorded signal. Ideally, a bandpass filter of 300–3000 Hz should be applied to reduce the influence of local field potentials on the recorded signal. In this study, the spike timestamps used to produce the activity maps (raster plots) were based on a standard thresholding system available through the Multichannel Systems recording software, which manually sets the threshold for each channel at the start of each recording (±5 s.d.). Although relatively common, this approach is not optimal as it does not account for potential signal-drift during the recording period. However, manual inspection of the spikes measured at each channel *post hoc* confirmed that the

recorded spikes tend to correspond to expected values of extracellularly measured action potentials (figure 5) as identified through the signal slope (spike shape) and voltage fluctuation.

Data accessibility. The electrophysiological raw-data files are available within Mendeley as Devold Valderhaug, Vibeke; Sandvig, Axel; Sandvig, Ioanna (2019), 'oeRecordings from 3D NSC-derived neural networks', http://dx.doi.org/10.17632/r9gd7g8zcy.4 [22] under a CC-BY licence. The MEA Analysis Toolbox is available for download at https://github.com/helgeanl/MEA_toolbox.

Authors' contributions. V.D.V. carried out the cell-based experiments and data collection, performed the analysis and wrote the paper; W.R.G. contributed to the conception and design of the study, performed analysis on the size distribution of the polymer particles and wrote parts of the Material and methods section; E.M.S. contributed with functionalization and size distribution analysis of the polymer particles, and contributed to the writing of the Material and methods section; M.Y. contributed to design and functionalization of the polymer particles; A.S. conceived and designed the study, and helped draft and critically revise the manuscript; I.S. conceived and designed the study, carried out the cell-based experiments, and helped draft and critically revise the manuscript. All authors gave final approval for publication and agree to be held accountable for the work performed therein.

Competing interests. We declare we have no competing interests.

Funding. This work was supported by the Department of Neuromedicine and Movement Science, Faculty of Medicine and Health Sciences, NTNU; The Liaison Committee for Education, Research and Innovation in Central Norway; the Joint Research Committee between St Olav's Hospital and the Faculty of Medicine and Health Sciences, NTNU; and internal strategic funding at SINTEF AS.

Acknowledgements. The scanning electron microscopy (SEM) preparation and imaging were provided by the Cellular and Molecular Imaging Core Facility (CMIC), Norwegian University of Science and Technology (NTNU). CMIC is funded by the Faculty of Medicine at NTNU and Central Norwegian Regional Health Authority. Activity maps (raster plots) were produced using an in-house Matlab toolbox created by Helge-André Langåker and Martinius Knudsen at the Department of Engineering Cybernetics at NTNU.

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
