## [Reviewer comments · Royal Society Open Science]

Review History

RSOS-191086.R0 (Original submission)

Review form: Reviewer 1

Is the manuscript scientifically sound in its present form?

Yes

Are the interpretations and conclusions justified by the results?

Yes

Is the language acceptable?

Yes

Do you have any ethical concerns with this paper?

No

Have you any concerns about statistical analyses in this paper?

No

Recommendation?

Accept with minor revision (please list in comments)

Comments to the Author(s)

Few minor comments

1-in the introduction, lines 26-33 are long and vague, it needs to be rewritten.

2- in the result section currently there is one subheading which is 3.1 (physiological measurements...). I suggest to include another subheading for the first part of the results (for example neural network culture...). This will make it clearer to the reader.

3- in the result section, line 46 there is a typo error "fuorescece", change fluorescence.

Review form: Reviewer 2

Is the manuscript scientifically sound in its present form?

Yes

Are the interpretations and conclusions justified by the results?

Yes

Is the language acceptable?

Yes

Do you have any ethical concerns with this paper?

No

Have you any concerns about statistical analyses in this paper?

No

Recommendation?

Accept with minor revision (please list in comments)

Comments to the Author(s)

The authors develop a novel 3D topology of neural networks using polymer particles. The need to do this is justified by stating that dimensionality reduction affects the nature of the represented information. However, dimensionality reduction is typically performed not by reference to geometric space, rather with respect to encoding high-dimensional stimulus spaces. I believe the purpose of the work, especially in the introduction and the abstract, needs to be better justified. Otherwise, it appears to be of interest and is a well written article.

Decision letter (RSOS-191086.R0)

22-Aug-2019

Dear Ms Valderhaug

On behalf of the Editors, I am pleased to inform you that your Manuscript RSOS-191086 entitled "Formation of neural networks with structural and functional features consistent with small-world network topology on surface-grafted polymer particles" has been accepted for publication in Royal Society Open Science subject to minor revision in accordance with the referee suggestions. Please find the referees' comments at the end of this email.

The reviewers and handling editors have recommended publication, but also suggest some minor revisions to your manuscript. Therefore, I invite you to respond to the comments and revise your manuscript.

- Ethics statement

- Data accessibility

If you wish to submit your supporting data or code to Dryad (<http://datadryad.org/>), or modify your current submission to dryad, please use the following link:
<http://datadryad.org/submit?journalID=RSOS&manu=RSOS-191086>

- Competing interests

- Authors' contributions

- Acknowledgements

- Funding statement

Because the schedule for publication is very tight, it is a condition of publication that you submit the revised version of your manuscript before 31-Aug-2019. Please note that the revision deadline will expire at 00.00am on this date. If you do not think you will be able to meet this date please let me know immediately.

- 1) A text file of the manuscript (tex, txt, rtf, docx or doc), references, tables (including captions) and figure captions. Do not upload a PDF as your "Main Document";
- 2) A separate electronic file of each figure (EPS or print-quality PDF preferred (either format should be produced directly from original creation package), or original software format);
- 3) Included a 100 word media summary of your paper when requested at submission. Please ensure you have entered correct contact details (email, institution and telephone) in your user account;
- 4) Included the raw data to support the claims made in your paper. You can either include your data as electronic supplementary material or upload to a repository and include the relevant doi

within your manuscript. Make sure it is clear in your data accessibility statement how the data can be accessed;

5) All supplementary materials accompanying an accepted article will be treated as in their final form. Note that the Royal Society will neither edit nor typeset supplementary material and it will be hosted as provided. Please ensure that the supplementary material includes the paper details where possible (authors, article title, journal name).

on behalf of Dr Derek Abbott (Associate Editor) and Pietro Cicuta (Subject Editor)
openscience@royalsociety.org

Reviewer comments to Author:
Reviewer: 1

Few minor comments

1- in the introduction, lines 26-33 are long and vague, it needs to be rewritten.

2- in the result section currently there is one subheading which is 3.1 (physiological measurements...). I suggest to include another subheading for the first part of the results (for example neural network culture...). This will make it clearer to the reader.

3- in the result section, line 46 there is a typo error "fuorescece", change fluorescence.

Reviewer: 2
Comments to the Author(s)

The authors develop a novel 3D topology of neural networks using polymer particles. The need to do this is justified by stating that dimensionality reduction affects the nature of the represented information. However, dimensionality reduction is typically performed not by reference to geometric space, rather with respect to encoding high-dimensional stimulus spaces. I believe the purpose of the work, especially in the introduction and the abstract, needs to be better justified. Otherwise, it appears to be of interest and is a well written article.

Author's Response to Decision Letter for (RSOS-191086.R0)

See Appendix A.

Decision letter (RSOS-191086.R1)

10-Sep-2019

Dear Ms Valderhaug,

I am pleased to inform you that your manuscript entitled "Formation of neural networks with structural and functional features consistent with small-world network topology on surface-grafted polymer particles" is now accepted for publication in Royal Society Open Science.

on behalf of Dr Derek Abbott (Associate Editor) and Pietro Cicuta (Subject Editor)
openscience@royalsociety.org

Associate Editor Comments to Author (Dr Derek Abbott):

Associate Editor: 1

Comments to the Author:

(There are no comments.)

Reviewer comments to Author:

Appendix A

Response to Referees

We would like to thank the editor and referees for their positive feedback and useful suggestions. A point-by-point response to the comments by referees can be found below. All relevant changes are highlighted in the manuscript file entitled “Revised Main Document, highlighted changes”.

Reviewer: 1

“1-in the introduction, lines 26-33 are long and vague, it needs to be rewritten.”

Response: Lines 26-33 have been rewritten and are now more concise

“2- in the result section currently there is one subheading which is 3.1 (physiological measurements...). I suggest to include another subheading for the first part of the results (for example neural network culture...). This will make it clearer to the reader.”

Response: A new subheading for the first part of the results has been added:
“3.1 Neural network culture”

“3- in the result section, line 46 there is a typo error "fuorescece", change fluorescence.”

Response: The typo has been corrected

Reviewer: 2

“The authors develop a novel 3D topology of neural networks using polymer particles. The need to do this is justified by stating that dimensionality reduction affects the nature of the represented information. However, dimensionality reduction is typically performed not by reference to geometric space, rather with respect to encoding high-dimensional stimulus spaces. I believe the purpose of the work, especially in the introduction and the abstract, needs to be better justified.

Otherwise, it appears to be of interest and is a well written article.”

Response: The abstract and a part of the introduction has been rewritten to better highlight the purpose of the work.